

# Enhancing secure multi-group data sharing through integration of IPFS and hyperledger fabric

Feng Wen[1], Zhuo Wang[1], Leda Qu[1], Haixin Huang[2] and Xiaojie Hu[1]

[1] School of Information Science and Engineering, Shenyang Ligong University, Shenyang, Liaoning, China
[2] School of Automation and Electrical Engineering, Shenyang Ligong University, Shenyang, Liaoning, China

## ABSTRACT

Data sharing is increasingly important across various industries. However, issues such as data integrity verification during sharing, encryption key leakage, and difficulty sharing data between different user groups have been identified. To address these challenges, this study proposes a multi-group data sharing network model based on Consortium Blockchain and IPFS for P2P sharing. This model uses a dynamic key encryption algorithm to provide secure data sharing, avoiding the problems associated with existing data transmission techniques such as key cracking or data leakage due to low security and reliability. Additionally, the model establishes an IPFS network for users within the group, allowing for the generation of data probes to verify data integrity, and the use of the Fabric network to record log information and probe data related to data operations and encryption. Data owners retain full control over access to their data to ensure privacy and security. The experimental results show that the system proposed in this study has wide applicability.

## INTRODUCTION

The Internet has revolutionized data access, making data sharing a ubiquitous aspect of daily life. However, sharing data over public networks exposes it to significant security risks, such as data error handing and tampering (*Anwar et al., 2022*). This has led to a major challenge in ensuring data integrity and verifiability. Trusted third parties play a crucial role in facilitating secure data sharing, as they can authenticate and reliably transmit data (*Naz et al., 2019*). Therefore, it is of utmost importance to develop data sharing techniques that guarantee both data integrity and verifiability, as well as reliable transmission.

Group data sharing is an essential requirement for various industries, where multiple users in a group share information for collaborative purposes. The complexity of such data sharing scenarios includes medical institutions wishing to share medical databases, schools wishing to share their research results, *etc* (*Huang, Chen & Wang, 2020*). Existing research approaches to implement data sharing between users in a group consist of one-to-many

Corresponding author
Xiaojie Hu,
xiaojie.hu.wmu@gmail.com

and many-to-many. The one-to-many approach is suitable when only one data owner is present, and authorized users can access their data resources. On the other hand, the many-to-many approach allows authorized users to access the data resources of different data owners within the same group. However, current research mainly focuses on a single scenario where all users within the group want to share data. In such scenarios, data is usually stored for sharing using cloud servers.

The current data transmission technology typically employs data encryption prior to transmission. The server uses a specific secret key to send the data to the client. The client then processes the data with the secret key, obtains the encrypted data, and transmits it to the server. Upon receiving the encrypted data, the server uses the same secret key to decrypt the data and restore it to its original form. However, this approach is vulnerable to data breaches during transmission due to the low security and reliability of data transmission. Therefore, it is necessary to explore more secure and reliable methods of data transmission to prevent data leakage.

In recent years, blockchain technology has been widely adopted to address the issues of data loss, tampering and leakage in data sharing. Blockchain's decentralized nature ensures that each block release is not controlled by a central node, thereby ensuring the integrity and traceability of data. Additionally, the security encryption algorithm of blockchain provides security and privacy for data. Any attempt to intrude and modify the data will be recorded and easily traceable. The distributed storage technology of blockchain further ensures that data loss is avoided. The decentralized, tamper-evident, and traceable features of blockchain provide the foundation for building a transparent, open, secure, and trustworthy data sharing environment that can connect big data in various fields.

Hyperledger Fabric is a popular open-source framework for hyperledger technology. It is a consortium blockchain open-source framework that enables multi-sector participation in data sharing operations, while its multi-module design allows users to customize the service according to their needs.

The InterPlanetary File System (IPFS) is an innovative distributed file storage system that utilizes P2P network, content addressing, and distributed hash table (DHT) technology to provide high levels of security and integrity. Peers in the IPFS network can exchange files efficiently without mutual trust.

The use of blockchain and IPFS in data sharing still presents some challenges. First, since the resource data is encrypted and shared on the blockchain, this leads to a lack of control over the data and increases the storage pressure and block generation time of the blockchain. Second, it is difficult to encrypt data during transmission in IPFS-based data sharing. Once the content identifier (CID) of the data is leaked, any user with the CID can share the data, rendering data privacy meaningless. Moreover, there is a risk of data interception during data transmission.

We propose a data sharing system based on IPFS and Hyperledger Fabric. All data in the system is stored locally by the data owner. When the data owner requests to upload data, a data probe is generated by uploading the data to IPFS. This probe is used only to detect data tampering or duplication and other users cannot download the data using its CID. The metadata of the data, along with its CID, are recorded in the Fabric ledger as a

data directory. The system uses Fabric's built-in private data feature to implement the sharing of data and the division of private data between groups. Data owners have complete control over data access and can authenticate permissions through Fabric's smart contract. When a user requests to download data, a dedicated P2P network is established between the data requester and owner to facilitate data transfer. During data transfer, the system generates dynamic keys based on the data transmission time to encrypt the data, ensuring secure data transfer and avoiding security risks from key leakage.

Our main contributions are summarized as follows:

(1) We design a data sharing model and a decentralized, transparent, and public interaction environment based on consortium blockchain and IPFS. The model enables reliable recording of important information during interactions, which are traceable and immutable. The environment allows for secure sharing of multiple groups of data between data owners and data requesters.

(2) We propose a scheme to verify data integrity using IPFS technology, which can realize automatic verification of integrity verification in data sharing.

(3) We present a P2P data transmission mode and a dynamic key generation algorithm. Data is encrypted during transmission between owners and requesters, and dynamic keys are generated to prevent key leakage and ensure data security.

## RELATED WORK

Blockchain technology, which originated from Bitcoin (*Nakamoto, 2008*), offers a decentralized approach for establishing trust relationships between anonymous parties while ensuring document privacy and security (*Nair & Dorai, 2021*). PDPChain (*Liang et al., 2022*) proposed a personal data privacy protection scheme based on consortium blockchain that stores original data encrypted with an improved Paillier homomorphic encryption mechanism, namely PDPChain, where users realize fine-grained access control based on ciphertext policy attribute-based encryption (CP-ABE) on blockchain. As the functionality of shared encrypted data is relatively constrained and the construction of the CP-ABE model is intricate, the method for constructing the access model is not detailed in this study. The integration of multi-party secure computing and blockchain facilitates privacy-preserving computation (*Zhou et al., 2021*). However, it is noteworthy that multi-party secure computing introduces a trade-off by exerting a certain impact on the system's performance. To address the issue of increasing storage costs associated with blockchain ledgers, several studies have proposed the use of IPFS to generate transaction hashes that replace original data in blocks with CIDs of the transactions (*Zheng et al., 2018*; *Kumar & Tripathi, 2019*). Recent research have focused on developing file management systems that utilize both IPFS and Hyperledger Fabric blockchain (*Huang et al., 2020*). For instance, BlockIPFS (*Nyaletey et al., 2019*) proposes an innovative approach to achieve traceability of file access through Hyperledger Fabric. FileWallet (*Chen et al., 2022*) proposes a peer-to-peer (P2P) file management system architecture based on IPFS and Hyperledger Fabric to support file content updates and direct file synchronization across devices. These studies primarily center on leveraging the IPFS for data sharing, which is a systematic storage framework founded on distributed files. However, utilization of a third-party file system to

store data poses certain threats to data security, necessitating efforts to fortify data integrity. This study proposes a novel approach for ensuring data integrity by leveraging the CID returned by IPFS as a probe. Specifically, the approach involves verifying data integrity by comparing the CID of the original data with the CID of the data retrieved from IPFS. The study demonstrates the effectiveness of this approach in verifying the integrity of both text and image data, and highlights its potential applications in fields such as copyright protection and data forensics.

Designing an appropriate data sharing scheme for group data sharing requires a tailored approach that takes into consideration the complex application scenarios. One potential solution, presented in a recent study (Shen et al., 2017a), is based on symmetric balanced incomplete block design (SBIBD) and group signature technique. This approach enables anonymous and secure group data sharing within the same group. Other studies propose privacy-preserving and untraceable data sharing schemes that utilize proxy re-encryption and Oblivious Random Access Memory (ORAM) to support multiple users in a group to share data in cloud computing (Shen et al., 2021). Such schemes hold promise for advancing group data sharing and facilitating secure collaborations among multiple parties.

To ensure secure data sharing, data integrity verification is a crucial security requirement (Chen et al., 2014; Zhang et al., 2019). Recently, research involve two solutions, the two-party auditing model and the third-party auditing model schemes. The two-party auditing model (Deswarte, Quisquater & Saidane, 2004; Gazzoni Filho & Barreto, 2006) is the first data auditing model used in data sharing schemes, where a separate auditing service was required for data integrity verification. However, with the advent of cloud servers and IPFS, these platforms have been leveraged for data storage and sharing. Subsequently, a third-party auditing model scheme, known as the provable data possession (PDP) model, was introduced for data integrity verification (Ateniese et al., 2007). This scheme incorporates an attribute-based multi-verifier cryptographic verification control algorithm for remotely outsourced data integrity verification (Xu et al., 2022). Despite their differences, both approaches rely on a third-party to verify data integrity. Wang, Zhang & Zhang (2019) proposed a new personal health records sharing scheme based on blockchain with data integrity verifiable. In particular, the new scheme in this study stores the hash values of encrypted personal health records in blockchain, and the related index set is stored in smart contract, which can further improve the efficiency of data integrity verification. This study advocates the utilization of smart contracts for the verification of data integrity. However, the storage of data *via* cloud servers poses a potential threat to data security. Furthermore, it is essential to note that this investigation relies on Ethernet implementation. Given that transactions in Ether necessitate the involvement of currency, the applicability of this approach may be limited in certain scenarios.

The encryption of data during transmission is a crucial aspect of secure communication. Research on encryption methods and key management has therefore gained significant attention. For instance, Shen et al. (2017b) proposed a key negotiation protocol based on block design, which utilizes SBIBD and group data sharing models to generate a public key

IC for multiple participants, enabling secure data sharing within and outside a group in cloud computing environments. *Capar et al. (2010)* implemented an elliptic curve-based DH key sharing method using fast frequency hopping to enhance the security of data transmission. *Wong, Shea & Wong (2008)* developed a reliable auto-repeating fast fading channel-based shared key generation method. Another proposed approach for secure communication in wireless networks involves generating keys using wireless transmission error randomness (*Xiao & Gong, 2012*). Dynamic secrets were generated by listening to data link layer data, albeit using a high communication overhead RSA encryption scheme for data encryption (*Xiao, Gong & Towsley, 2010*; *Sun et al., 2012*). Similarly, *Liu et al. (2013)* proposed a dynamic key scheme using a 0–1 retransmission sequence to compute a private shared key, thereby reducing communication overhead. However, this approach is reliant on network environments and poses a risk of key leakage. In addition to symmetric encryption algorithms like DES and AES, researchers also use asymmetric encryption algorithms, such as homomorphic encryption for secure aggregation (*Iyer, 2011*) and multiparty secure computation (*Saputro & Akkaya, 2012*). However, these methods only perform specific and limited operations on encrypted data and are computationally intensive, thus making them unsuitable for various types of data transmission. Conversely, dynamic keys have the advantage of being used only once, and by designing dynamic keys and encrypting files based on transmission time, the method obviates the need for key transmission and storage.

# METHOD

This section presents the model architecture of data sharing, including the design of data structure, smart contract, and user authentication function, as well as the functional modules and security analysis of the system. Specifically, we describe the system's data storage and access function, and how the smart contract is designed to facilitate the storage and access of the data directory. We also provide detailed insights into the implementation of user access control through smart contracts, ensuring a robust framework for secure access to shared data. Additionally, we provide an in-depth analysis of the security of the system, and the design of its functional modules to achieve efficient and reliable data sharing.

## The model architecture

The system comprises two main components, namely the data sharing system and group data management system, see Fig. 1. The data sharing system facilitates communication with the Fabric network to access the data sharing directory. The group data management systems respond to data download requests, generates dynamic keys for data encryption, and transfers data to the requester *via* a peer-to-peer (P2P) network. Data owner A utilizes the data sharing front-end processor within the data sharing system to update the data directory. The data directory can be stored in either a public or private state, with a ledger maintained by the group nodes. To monitor changes in the shared data, data probes are generated and recorded in the chain, ensuring their immutability. Data requester B can be a node from the same group as data sharing user A or from another group. The data

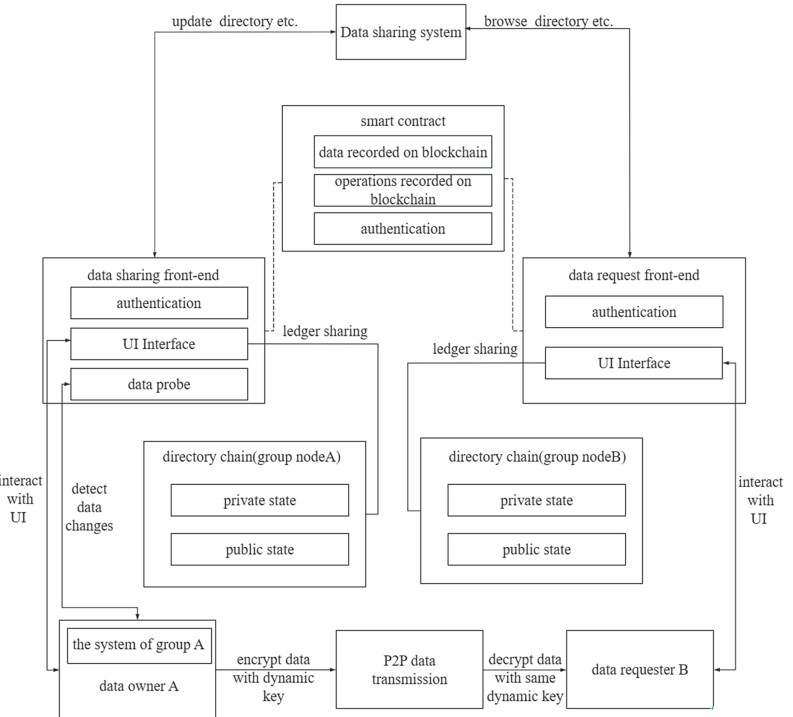

**Figure 1 The system architecture.** 

directory within the data sharing system is accessed through the request front-end processor. In this data sharing platform, authentication of all group nodes is required. To achieve autonomous authentication of permissions, this research employs smart contracts. These smart contracts are pre-installed on the departmental nodes and require endorsement for modification permissions. Consequently, the authentication of privileges through smart contracts ensures system security.

The system network structure and transaction proposal process are illustrated in Fig. 2. In our network structure, the smallest unit is represented by a node, which corresponds to an individual user. Users can assume the roles of either data owners or data requesters, depending on whether they are sharing or requesting data. User groups can be created, resembling real-world entities such as an academic department or an information management department within a school. Figure 2 demonstrates two distinct groups, A and B, comprising different users. The affiliation of data owners and data requesters within the same or different groups does not hinder the sharing of data among users. In the Hyperledger Fabric network, each participating node must have a certificate issued by the CA structure to join the network. Each organization in the Fabric is established as an independent group with a dedicated IPFS network based on its members. When a user initiates a data sharing request, the data is first uploaded to the group IPFS network, and the CID of the data is returned by IPFS as a unique identifier for the data. The CID is then stored in the Fabric ledger along with the data directory and other relevant information. When the data requester obtains the data from the data owner, the CID of the data can be calculated and compared with the information recorded in Fabric. The data probe serves to

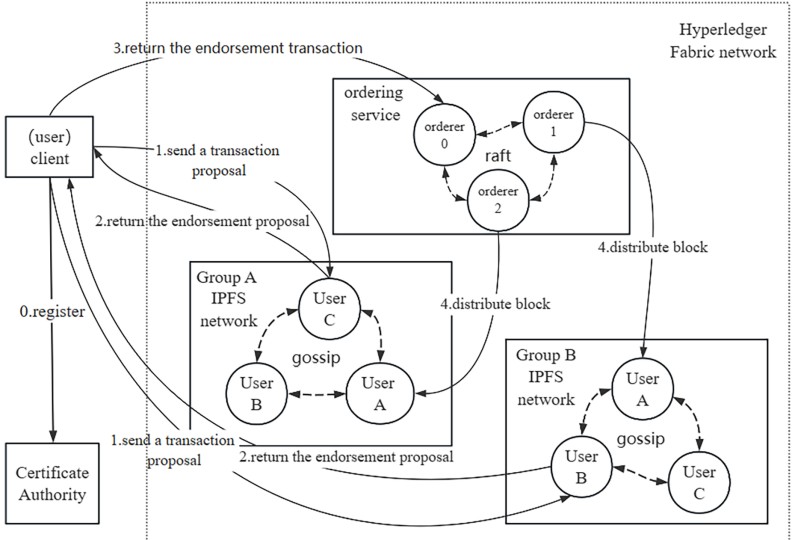

**Figure 2 System network structure.**

detect whether the data has been tampered with or duplicated during the sharing process. owner, the CID of the data can be calculated and compared with the information recorded in the Fabric, and the data probe can be used to detect the data whether tampering or repeated sharing occurred during the sharing process.

Figure 2 illustrates the transaction proposal process, commencing with the registration of the client node with the certificate authority (CA) to procure a certificate authorizing its integration into the Hyperledger Fabric network. Subsequently, the user-initiated transaction proposal is distributed to each group, where group nodes endorse and return the proposal to the client node. The endorsed proposal is then forwarded to the sorting node. Utilizing the Raft consensus protocol, the sorting node achieves consensus and organizes the proposal into a block. Finally, the newly formed block undergoes dissemination to each group, and nodes within the group leverage the gossip protocol to ensure uniform ledger data across users. In contrast to the Kafka consensus protocol, the Raft protocol is embedded in the order node, offering strong consistency and enhanced data consistency guarantees. Consequently, this research adopts the Raft consensus protocol to achieve consistent consensus within the blockchain network.

## Data structure

The default database used by Hyperledger Fabric, Leveldb, and the optional Couchdb are both non-relational databases that store data in key-value pairs. Storing data in JSON format is a convenient way to interact with these databases.

### User profile

The implementation of data sharing operations by users involves the connection to both the Hyperledger Fabric network and IPFS network nodes. The user's password is a hash value generated based on a certificate issued by the group CA authority. The user's Membership Service Provider (MSP) identifies the group to which the user belongs.

**Table 1  Data structure of user profile.**

| Name | Type | Description |
| --- | --- | --- |
| Id | String | The ID of the user. |
| Name | String | The name of the user. |
| Password | String | The keyword of the user. |
| Msp | String | The group tag of the user. |

**Table 2  Data structure of data directory.**

| Name | Type | Description |
| --- | --- | --- |
| Key | String | The key of the data directory. |
| Description | String | The description of the data. |
| dataAddress | String | The storage location of the data. |
| accessPolicy | Array | The access policy of the data. |
| Owner | String | The ip address of the data owner. |
| Uptime | String | The creation timestamp of data directory. |
| Cooperator | Array | The array of the cooperator name. |
| Uploader | Array | The array of the uploader name. |
| Downloader | Array | The array of the downloader name. |
| CID | String | The CID of the data |

During connection to the Fabric network, the hash value of the certificate is calculated to verify the user's identity. Table 1 shows the data structure of the user.

### Data directory

To address the challenge of cumbersome data upload procedures associated with substantial data capacity during data sharing, a targeted solution involves replacing specific data with a data directory. This directory incorporates detailed description such as data content, storage path, access policies, data ownership, and a unique identifier (CID). Furthermore, to achieve nuanced control over data permissions, the delineation includes collaborators (empowered for collaborative data editing), downloaders (authorized to download shared data), and uploaders (enabled to upload shared data). This approach affords users comprehensive control over the shared data during the process of data sharing, ensuring a fine-grained division of permissions. The data structure of data directory is shown in Table 2. The data directory records the metadata of shared data.

The access policy mainly records the groups that have permission to share the data. Collaborators of data refer to all users who have data collaboration privileges, allowing them to modify and delete the content of the shared data. Uploaders of data refer to all users who have data sharing privileges and can publish shared data directories. Downloaders of data refer to all users with data download permission, enabling them to download shared data. Data sharers control access to the shared data and can modify the data collaborators and downloaders of the shared data.

**Table 3 Smart contract functions.**

| Function name | Description |
|---|---|
| CreateNewdirectory | Create a shared data directory in ledger. |
| updateDirectory | Update a shared data directory in ledger. |
| deleteDirectory | Remove a shared data directory in ledger. |
| CreateNewprivatedirectory | Create a private data directory in ledger. |
| updatePrivateDirectory | Update a private data directory in ledger. |
| deletePrivateDirectory | Remove a private data directory in ledger. |
| updateCooperator | Update cooperators in a data directory. |
| updateDownload | Update downloaders in a data directory. |
| updateUpload | Update uploaders in a data directory. |
| createUploader | Create a data directory to store the uploaders. |
| varifyDownload | Verify whether the user can download specific data. |

## Smart contract design

Serving as a pivotal interface for blockchain interactions, smart contracts inherently possess attributes of automated execution and irrevocability, rendering them advantageous for the automated storage of data directory and user access restriction. In light of these characteristics, we have devised an access control function founded on smart contracts, facilitating the effective implementation of user access control. Additionally, to discern between data catalogs in private and public states within the Hyperledger Fabric environment, we have tailored smart contracts capable of distinct operations on private and public data directory. The delineation of the smart contract interface is shown in Table 3.

First six smart contracts in Table 3, each of which is accompanied by a detailed algorithmic flow described in the subsequent section. Among these contracts, "updateCooperator" and "updateDownload" are employed to update the sets of collaborators and downloaders associated with the shared data directory, respectively. The "createUploader" contract is executed during the initialization of the blockchain network and serves the purpose of recording users with permission to share the data directory. Additionally, the "updateUpload" contract is utilized to modify the set of data uploaders. Lastly, the "varifyDownload" contract is responsible for verifying a user's eligibility to download data.

## Functional module design

This section presents the primary functional modules of the system, which consist of shared data directory management, private data directory management, data probe management, and data sharing.

### Public data directory management

This module is responsible for managing new public state data, which is visible to all groups. Firstly, the module verifies if the user has the uploader permission, and only

---

**Algorithm 1** CreateNewDirectory.

**Input:** Data metadata $data_{meta}$, data directory key $k$

**Output:** Excution results $result$

//Get the current user information of the system

1  $Uploader \leftarrow$ getUserName();

2  $T \leftarrow$ getTimestamp();

3  //Create a new data directory

4  $data\ directory \leftarrow$ newDirectory($data_{meta}$,$Uploader$,$T$);

5  **if** *The Uploader has upload privilege and file has not been shared* **then**

6      //The data directory is recorded on the public ledger in the blockchain

7      putStringState ($k$,$data\ directory$);

8       **return** $true$;

9  **else**

10    Throw privilege error;

11 **end**

---

**Algorithm 2** updateDirectory.

**Input:** Data directory key $k$,data metadata $data_{meta}$

**Output:** Excution results $result$

//Get the data directory from the public ledger according to the $k$

1  $data\ directory \leftarrow$ getStringState($k$);

2  //Get the current user information of the system

3  $user \leftarrow$ getUserName();

4  **if** *The user has cooperator privilege* **then**

5          $new\ data\ directory \leftarrow$ updateDirectory($data_{meta}$,$user$);

6          putStringState($k$,$new\ data\ directory$);

7          **return** $true$;

8  **else**

9          Throw privilege error;

10   **end**

---

authorized users can share the data. Secondly, when modifying the public state data, it verifies if the user has collaborator privileges. Users with the appropriate permissions can modify the data in the public state. Finally, when deleting data from the public state, the module checks whether the authenticated user has collaborator privileges. Only users with the appropriate permissions can delete data from the public state. Algorithms 1–3 illustrate the procedure of shared data directory management.

**Algorithm 3** deleteDirectory.

**Input:** Data directory key $k$

**Output:** Excution results *result*

//Get the data directory from the public ledger according to the $k$

1    *data directory* ← getStringState($k$);

2    **if** *The user has cooperator privilege* **then**

3        //Delete the data directory from the public ledger according to the $k$

4        delState($k$);

5        **return** *true*;

6    **else**

7        Throw privilege error;

8    **end**

### Group private data directory management

This module is designed to manage private state data that is only visible within a specific group. When adding new data to the private state, users must specify the access policy as limited to a particular group. The functional module is similar to the shared data directory management module, with the additional requirement that users specify the name of the private state for the operation. Furthermore, the user's group and the privileges are authenticated during privilege authentication. Algorithms 4–6 illustrate the procedure of private data directory management.

### Data probe management

The data probe management module is responsible for ensuring the integrity and uniqueness of shared data. To accomplish this, the system generates data probes that detect data tampering and duplication. The probe generation function uploads the data directory to the IPFS private network and returns the CID of the data as the data probe. During data retrieval, the detection function calculates the CID of the data and compares it with the CID of the data in the Fabric ledger. If the CIDs match, the data is considered authentic. Furthermore, the probe screen for duplicates function checks for duplicate shared resources. To this end, the data directory generates data CIDs through IPFS before recording them in the Fabric ledger. These CIDs are then compared with the set of CIDs in the ledger. If there is a match, it means that the data has already been shared, and the system will reject the user's data sharing request. Algorithms 7–9 illustrate the procedure of data probe management.

### Data sharing

Key leakage is a serious threat to data transmission security, and dynamic keys are essential to ensure secure data transmission. Two common types of dynamic keys are counted-use and timed-use OTPs. Counted-use OTPs can be used indefinitely, and a new password is generated after each successful use by adding 1 to the counter. Time-based OTPs, on the

---

**Algorithm 4  CreateNewprivatedirectory.**

**Input:** data metadata $data_{meta}$,group name $name_{group}$, data directory key $k$

**Output:** Excution results $result$

//Get the current user information of the system

1     $Uploader \leftarrow$ getUserName();

2     //Gets the group MSP name of the user

3     $group\ Msp \leftarrow$ getUserMsp();

4     $T \leftarrow$ getTimestamp();

5     $data\ directory \leftarrow$ newDirectory($data_{meta}$,$Uploader$,$T$);

6     **if** *The Uploader has upload privilege and belongs to this group and file has not been shared* **then**

7          //The data directory is recorded on the private ledger of the group

8          putPrivateData($k$,$name_{group}$,$data\ directory$);

9          **return** *true*;

10    **else**

11         Throw privilege error;

12    **end**

---

**Algorithm 5  updatePrivateDirectory.**

**Input:** Data directory key $k$,data metadata $data_{meta}$,group name $name_{group}$

**Output:** Excution results $result$

1     $user \leftarrow$ getUserName();

2     //Gets the group name of the user

3     $group\ Msp \leftarrow$ getUserMsp();

4     **if** *The user belongs to this group* **then**

5          //Get the *data directory* from the private ledger of the group according to the $k$

6          $data\ directory \leftarrow$ getPrivateDataUTF8($name_{group}$,$k$);

7     **else**

8          Throw privilege error;

9     **end**

10    **if** *The user has cooperator privilege* **then**

11         $new\ data\ directory \leftarrow$ updateDirectory($data_{meta}$,$user$);

12         putPrivateData($name_{group}$,$k$,$new\ data\ directory$);

13         **return** *true*;

14    **else**

15         Throw privilege error;

16    **end**

---

 

**Algorithm 6** deletePrivateDirectory.

    **Input:** Data directory key $k$, group name $name_{group}$

    **Output:** Excution results $result$

1    $user \leftarrow$ getUserName();

2    $group\ Msp \leftarrow$ getUserMsp();

3    **if** *The user belongs to this group* **then**

4        *data directory* $\leftarrow$ getPrivateDataUTF8($name_{group}$,$k$);

5    **else**

6        Throw privilege error;

7    **end**

8    **if** *The user has cooperator privilege* **then**

9        //Delete the *data directory* from the private ledger according to the $k$

10        delPrivateData($name_{group}$,$k$);

11        **return** *true*;

12    **else**

13        Throw privilege error;

14    **end**

---

**Algorithm 7** CreateProbe.

    **Input:** Upload data $data_{upload}$

    **Output:** data probe $CID$

1    **if** *The user has upload privilege and file has not been shared* **then**

2        $CID \leftarrow$ IPFSUpload($data_{upload}$) and add the CID to the blockchain network;

3        **return** $CID$;

4    **else**

5        Throw privilege error;

6    **end**

---

other hand, use HMAC-Based One-Time Password Algorithm (HOTP) to implement the dynamic password, and can set the validity period of the password between 30 s and 2 min. Another algorithm used for time-based dynamic passwords is the Time-based One-time Password (TOTP). The authentication principle for dynamic passwords involves sharing a key, also known as a seed key, between the authenticating parties. The algorithm then calculates the password using the same seed key for a particular event count or time value. The algorithms used for this purpose are symmetric algorithms such as HASH, HMAC, and others. These algorithms serve as the foundation for implementing all dynamic cryptographic algorithms.

---

**Algorithm 8 ProbeTamperproof.**

    **Input:** Download data $data_{download}$, data directory key $k$

    **Output:** Verification results *result*

    *CID* ← getStringState($k$);

    *new CID* ← IPFSUpload($data_{download}$);

1    **if:** *CID == new CID* **then**

2        **return** *true*;

3    **else**

4        **return** *false*;

5    **end**

---

**Algorithm 9 ProbeDuplicateproof.**

    **Input:** upload data $data_{upload}$

    **Output:** Verification results *result*

    *CID* ← IPFSUpload($data_{upload}$);

1    **if** *CID already exists in blockchain network* **then**

2        Throw Duplicate sharing error;

3    **else**

4        you can add the *CID* to the blockchain network and upload data directory;

5        **return** *true*;

6    **end**

---

The generation of dynamic keys is dependent on both a static key K and a random number C. The design of C is crucial to implementing dynamic keys. The HOTP algorithm is a one-time cipher generation algorithm based on event counting, with its shift factor C being a binary representation of the counter's value. Conversely, TOTP utilizes the current timestamp as the time difference, which is divided by the time window (default of 30 s) to obtain the time window count, which is used as the movement factor C for the dynamic cryptographic algorithm. Therefore, the design of a time-based dynamic key algorithm can be achieved to ensure secure transmission of files during data transmission. The process of dynamic key algorithm TOTP256 is shown in Algorithm 10:

The data sharing module consists of three components: data directory uploading, data collaboration sharing, and data downloading. Users must be authenticated before uploading a data sharing directory. If they have permission to share data, the data directory can be added to the Fabric ledger. The access policy determines whether the data directory is shared or private. Data collaborators can update shared data, while data downloaders can only download data. When a data requester sends a data download request, the system verifies their download or collaborator permission. The data owner then responds to the

---

**Algorithm 10** TOTP256.

**Input:** Static Secret $secret_{static}$, Timestamp $T$, key length $length$

**Output:** Dynamic Secret $secret_{dynamic}$

1   X ← 30;

2   T ← T/X;

3   //Generate a 16-digit string based on the timestamp

4   steps ← Long.toHexString(T).toUpperCase();

5   **if** $steps.length$ < 16 **then**

6       steps ← "0" + steps;

7   **end**

8   //Generate key using HmacSHA256 encryption algorithm

9   hmac ← Mac.getInstance("HmacSHA256");

10  macKey ← new SecretKeySpec($secret_{static}$, "RAW");

11  hmac.init(macKey);

12  hmackey ← hmac.doFinal(steps);

13  //Intercept the dynamic key of fixed length $length$

14  $secret_{dynamic}$ ← new String(hmackey,StandardCharsets.UTF_8);

15  **if** $secret_{dynamic}$.length < $length$

16      $secret_{dynamic}$ ← "0" + $secret_{dynamic}$;

17  **end**

18  **return** $secret_{dynamic}$;

---

data download request. If the download request is granted, a P2P data transfer channel is established between the two parties through sockets. The data owner generates a dynamic key based on the agreed point in time and encrypts the data during transmission. The data requester generates a dynamic key based on the same point in time when receiving the data and decrypts the received data. The socket connection is terminated at the end of the data transfer to complete the data download. The network transmission procedure for data sharing is illustrated in Fig. 3. Algorithm 11 illustrates the procedure of data download management.

## Analysis

This section provides an overview of the analytics system, which includes data security, data privacy, and data record traceability features.

### Security and privacy

The security and privacy of the system are crucial for the success of the analytics system. The reliability of Hyperledger Fabric is a key factor in ensuring the security of the system. The Hyperledger Fabric network is a permissioned blockchain, and all nodes that join the network must be licensed and issued certificates by a CA authority. If an organization does

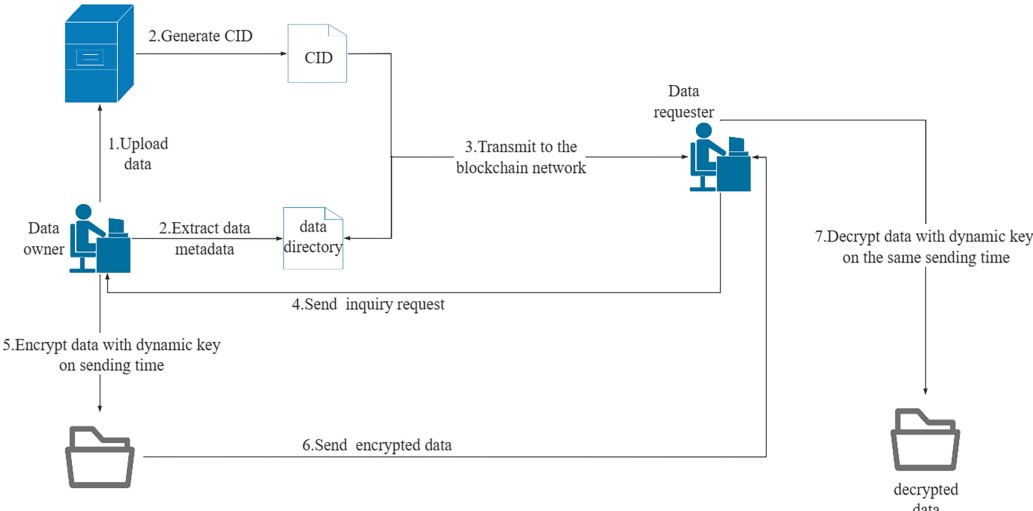

**Figure 3** **Network transmission mode of data sharing.**

---

**Algorithm 11** **dataDownload.**

**Input:** Data directory key *k*

**Output:** Execution results *result*

1    **if** *The user has download privilege* **then**

2        *data directory* ← getStringState(*k*);

3        *ipAddress,port* ← get from the data directory;

4        //Establish a socket connection with the data recipient and the data owner

5        *socket* ← new Socket(InetAddress.getByName(*ipAddress*),*port*);

6        send data download request to the data owner;

7    **else**

8        Throw privilege error;

9    **end**

10   **if** *The data owner agree the request* **then**

11       *T* ← getTimestamp();

12       //Generate a dynamic key with the immutable Static secret and timestamp T

13       *dynamic secret* ← generateTOTP256(*Static secret,T,keylength*);

14       //Encrypt data with the dynamic secret

15       *Encrypted data* ← encry(*dynamic secret,data*);

16       and the data recipient decrypts data according to the same sending time

17       //close the socket connection

18       socket.close();

19       **return** *true*;

20   **else**

21       **return** *false*;

22   **end**

---

**Table 4 Comparison results of similar schemes.**

| Features | Ours | Pro. (Liang et al., 2022) | Pro. (Zhou et al., 2021) | Pro. (Xu et al., 2022) | Pro. (Wang, Zhang & Zhang, 2019) |
|---|---|---|---|---|---|
| Data integrity verification | Yes | No | No | Yes | Yes |
| TAMPER-PROOF | Yes | Yes | Yes | Yes | Yes |
| Data security | Yes | Yes | Yes | No | No |
| Dynamic key | Yes | No | No | No | No |
| IPFS | Yes | Yes | No | No | No |
| Data cloud storage | No | No | No | Yes | Yes |

not trust a user, the organization administrator can reject the user from joining the Fabric network and revoke the user's certificate at the same time. User certificates also need to be renewed on time to ensure the reliability of the Hyperledger Fabric network, which can guarantee the security of the data.

To protect the privacy of the data, the data is not directly uploaded to the Fabric ledger but rather the metadata of the data is recorded on the ledger as a data directory. This approach ensures that the privacy of the data is not compromised.

Additionally, the data is uploaded to the IPFS network without informing others, and the IPFS private network strictly controls the nodes that join the network. Therefore, even if the CID is leaked, others cannot access the data based on the data CID. The only way for users to access the shared data is to initiate a data request to the data owner and get the data owner's approval for the request. Thus, the system can ensure both data privacy and security. Furthermore, the schemes are subject to comparative analysis across various dimensions, encompassing data integrity verification, resistance to tampering, data security, and the utilization of dynamic key and IPFS-based technology (Liang et al., 2022; Zhou et al., 2021; Xu et al., 2022; Wang, Zhang & Zhang, 2019). The outcomes of the security analysis are presented in Table 4.

## Traceability

Traceability is a fundamental characteristic of blockchain technology that ensures transparency and accountability in the system. In this analytics system, users can easily view the historical data of the ledger by accessing the ledger data. The traceability feature allows users to track the modifications made to the ledger data, which ensures transparency and helps to prevent fraudulent activities. Users can view the historical version of the data directory, the uploader, and the historical CID of the data, which helps to identify any changes made to the data. This traceability feature is essential for data record keeping and provides an added layer of security to the system.

## Data integrity

This study advocates the adoption of the IPFS to securely store the hash values of shared data in a permanent manner. The hash values, once stored within the blockchain, remain immune to tampering and offer traceability, thereby safeguarding the integrity of the associated data. Subsequent to the successful download by the data requester, the system

autonomously generates the hash value for the acquired data and conducts a verification process. This rigorous verification mechanism serves to guarantee that the downloaded data remains immune to malicious tampering and inadvertent processing errors during the sharing process.

### Trust model

The security model of a smart contract's access control is contingent upon the endorsement policy established during the installation of the chain code. In adherence to the trust model, where tolerance extends to a maximum of n malicious nodes, the endorsement policy for the smart contract can be configured to require validation from n +1 endorsing peers. Consequently, this strategic setting ensures that transaction proposals instigated by potentially malicious nodes are unable to garner the requisite endorsements, fortifying the robustness of the access control mechanism.

### Resistance to ciphertext only attack

This study advocates the adoption of a dynamic key algorithm to generate keys during data transmission. The encryption of data involves the utilization of distinct keys for each transmission event, ensuring the generation of a new key on every data transmission instance. This innovative approach effectively mitigates the security concern of attackers attempting to recover plaintext or decipher the key from encrypted ciphertext, a vulnerability associated with the repeated use of the same encryption key.

## EXPERIMENT AND RESULTS

This section presents the implementation of the system, including the environment setup and network deployment.

### Environment settings

Table 5 presents the versions of the dependencies required for the development installation.

### Network deployment

#### Certificates generation

In order to join the Hyperledger Fabric network, certificates play a vital role in the authentication and establishment of TLS channels for secure communication between users. Fabric employs two types of certificates, namely MSP certificates for authentication and TLS certificates to prevent attacks during user communication.

To generate the necessary certificates for the network, the system uses the Fabric binary package's cryptogen tool. This process involves customizing the network node configuration based on the template configuration file and defining the administrators and users of the network group. Once the network configuration is defined, the certificates can be automatically generated through a command.

#### Order and peer deployment

Hyperledger Fabric is designed to be modular, allowing sorting services and organizing peer nodes to be deployed on different machines to reduce communication pressure on a

**Table 5 The version of dependencies in our implementation environment.**

| Dependencies | Version |
|---|---|
| Go | 1.18.3 |
| Hyperledger fabric | 2.4.1 |
| Docker engine | 1.23.2 |
| Docker-compose | 20.10.14 |
| IPFS | 0.17.0 |

single machine. Prior to deploying nodes, it is necessary to generate channel files and write configuration files to define relevant configuration and topology information across the network. The configtxgen tool is then used to generate Genesis blocks and channel files based on the configuration file.

In addition, this module writes yaml configuration files for the network order nodes and peer nodes based on the network topology, and uses docker-compose to automatically pull the required images and start the nodes. The channel file is used to generate the network channel and organize the nodes to join the channel through the creation block.

### Smart contract deployment

Smart contracts need to be deployed after the Hyperledger Fabric network. In this research, we use Java to write the smart contracts. Once the smart contract is written, it is packaged into a JAR file and uploaded to the peer node machine. This approach saves time during chain code installation. The chain code needs to be installed on each peer node of the group, and the group's administrator needs to approve the installation to join the channel. After all groups grant permission, one of the group's administrators needs to commit the chain code to the channel. Following this, Hyperledger Fabric can respond to client requests to the smart contract.

### IPFS deployment

The IPFS network plays a crucial role in the system. To begin with, it is essential to ensure that all system users have IPFS installed. Users need to interact with the IPFS nodes through the IPFS API and configure the API network IP and port number. The IPFS nodes are started using the "ipfs daemon" command, after which all nodes joining the network must be authorized. To implement a dedicated IPFS network within the group, users in the same group are configured, while users from other groups are denied access to this network.

### System implementation

As both Hyperledger Fabric and IPFS offer Java SDKs, this system is primarily developed using Java, with a lightweight front-end framework called Layui for the user interface. To interact with the Hyperledger Fabric and IPFS networks, users must first install IPFS and obtain the necessary certificates. The system consists of shared data management, private data management, shared data upload, shared data download, and private data download

pages. Additionally, the data transfer system is implemented using Java sockets. To fulfill a data sharing request, the user initiating the request must run this system to transfer the requested data.

# EVALUATION

The present study conducted an evaluation of the system, which comprised a sorting service deployed on three order nodes on one server, as well as an organization deployed on two servers. Each organization represented a group and included three peer nodes. All three servers ran on the Centos7 system, while the client was a Windows server.

## Functional evaluation

### Public data directory management

The public data directory management function is evaluated by the following cases.

1. A user with data sharing privileges uploads a shared data directory. The access policy selects two groups, and the evaluation result is whether the user can find the newly shared data directory in the shared data directory page.
2. Users without data sharing privileges upload the shared data directory. The evaluation result is that data sharing fails and there is no new shared data directory on the Shared Data Directory page.
3. A user with collaborator privileges updates the data directory. The evaluation result is that the updated data directory can be found on the Shared Data Directory page.
4. Users without collaborator privileges update the data directory. The evaluation results in a message that no permission to modify the data directory appears on the page.

### Group private data directory management

This part is evaluated in terms of whether different group users can see the group private data directory. The functionality is evaluated using the following cases.

1. A user with data sharing privileges uploads a private data directory. The access policy selects a single group and evaluates whether the newly added data directory can be found on the Private Data directory page.
2. A user without data sharing privileges uploads a private data directory. Evaluate whether data sharing fails and no new data directories are added to the Private Data directory page.
3. The private data directory page is accessed after logging in users from two different groups. Evaluate whether the data directories are different for private data, and the newly added private data directories are only displayed in one group page.

### Data probe management

This section tests whether the probe can detect data tampering and duplicate data sharing. The functionality is evaluated by the following cases.

1. Transferring data that is different from when the system is shared. The evaluation uses probe detection to find whether data tampering has occurred.
2. Sharing data that has not been shared before. Evaluate whether the data CID is generated and the shared data directory is uploaded to the Fabric ledger without errors.
3. Share data that has already been shared. Evaluate if a data CID is generated and prompt for no duplicate sharing if the data was already shared at the time of upload.

### Data sharing

This part evaluates the core modules of the framework, mainly to test whether the data can be encrypted, decrypted and transmitted properly. And test the data collaborator and downloader permission management. The following cases are used to evaluate the function.

1. A user without data downloader permission downloads data. Test whether it prompts data downloading failure and indicates that there is no data downloading permission.
2. A data sharer updates the set of data downloader and collaborator permissions. Test whether the permission can be updated and the shared data directory page shows the updated content.
3. Non-data sharers update the set of data downloader and collaborator permissions. Test if the update fails and indicates that there is no permission to modify the collaborator and downloader collections.
4. The data owner sends data without invoking the dynamic key algorithm by the data requester. Test if the data format is unrecognizable characters that received by the data requestor.
5. The data owner sends the data and the data requestor invokes the dynamic key algorithm. Test whether the data requestor can receive the shared data accurately.

## Data sharing efficiency evaluation

The system's performance is evaluated by generating data shares of various sizes to assess the efficiency of data sharing. The initial evaluation involves generating 10 data with sizes ranging from 100 to 1,000 MB. The results of the probe generation time and the time required for verifying data integrity during data sharing are presented. Table 6 captures the temporal demands associated with recording a transaction involving data directory storage. The findings reveal consistent transaction times across varied data sizes. Notably, the primary influencers on transaction time are network-related factors, encompassing network latency and bandwidth. Activities involving data probe generation and data integrity verification necessitate hash value computations, and the computational duration exhibits an incremental trend with expanding data sizes. Users initiate probes by initially computing the hash value of the data. Consequently, the timeframe dedicated to data integrity verification during testing is observed to be comparatively less than the time allocated for probe generation.

**Table 6 Time usage with different file sizes.**

| File size (MB) | Transaction (s) | Data probe generation (s) | Verify data integrity (s) |
|---|---|---|---|
| 100 | 0.083 | 0.838 | 0.521 |
| 200 | 0.073 | 2.923 | 1.255 |
| 300 | 0.111 | 3.246 | 1.772 |
| 400 | 0.083 | 3.854 | 2.186 |
| 500 | 0.102 | 4.263 | 2.585 |
| 600 | 0.094 | 4.932 | 3.461 |
| 700 | 0.086 | 5.547 | 3.725 |
| 800 | 0.093 | 6.783 | 4.204 |
| 900 | 0.109 | 7.416 | 4.655 |
| 1,000 | 0.103 | 8.132 | 5.012 |

**Table 7 Dynamic key generation time is independent of file sizes.**

| File size (MB) | Dynamic key generation (s) | Dynamic key |
|---|---|---|
| 100 | 0.52 | 02611078 |
| 200 | 0.504 | 35820769 |
| 300 | 0.5 | 39365934 |
| 400 | 0.515 | 29849760 |
| 500 | 0.501 | 99935938 |

The second evaluation assessed the efficacy of the dynamic key algorithm and the associated time required for dynamic key generation. The testing involved five datasets ranging in size from 100 to 500 megabytes. As depicted in Table 7, the dynamic key generation consistently requires approximately 0.5 s, demonstrating its efficiency irrespective of variations in data size. Notably, the results indicate that the dynamic key generation time remains independent of data size fluctuations. Furthermore, the dynamic key generation process proves successful, producing keys with a consistent length of 8 bits, aligning with the predetermined setting in the algorithm.

The outcomes of evaluations indicate that the transaction time associated with sharing data in the proposed scheme remains unswayed by variations in data size. Enhanced deployment of peer nodes enables the equitable distribution of endorsement workload, consequently reducing transaction times. Notably, the dynamic key generation time exhibits independence from data size, with each iteration yielding distinct keys. Elevating the complexity of the dynamic key can be achieved by adjusting its length, thereby enhancing the security of data transmission.

## CONCLUSION AND FUTURE WORK

In this study, we present a novel architecture for data sharing based on IPFS and Consortium Blockchain to address the issues of data tampering and loss of encryption keys. This architecture eliminates the need for central servers or uploading data to IPFS,

enabling users to store data locally without risking privacy breaches. Additionally, we design an access control method based on smart contracts to achieve fine-grained access control of shared data. Furthermore, the use of dynamic keys resolves the issue of key transmission and storage. These features make our model architecture highly applicable for secure and efficient data sharing.

However, there are still some shortcomings in this article. There are several aspects that could be explored for further research. For instance, we will explore the integration of CP-ABE within smart contracts to fortify access control mechanisms. Additionally, the investigation aims to incorporate advanced privacy-preserving techniques, such as homomorphic encryption and federated learning, to augment the privacy computation of the system. Moreover, due to the nature of IPFS generating CIDs based on data content, the CIDs generated for similar data content can vary greatly, making it difficult for the system to identify whether similar data is already shared or not. Lastly, data can only be transferred when the user is online, and data sharing is disabled when the user is offline, which can be inconvenient for users. Future research could address these issues to improve the functionality and security of the system.

### Funding
This work was supported by the 2020 Program for Liaoning Excellent Talents (LNET) in University (1100003000301), and the National Key Research and Development Projects (2022YFC3302500). The funders had no role in study design, data collection and analysis, decision to publish, or preparation of the manuscript.

### Grant Disclosures
The following grant information was disclosed by the authors:
2020 Program for Liaoning Excellent Talents (LNET) in University: 1100003000301.
National Key Research and Development Projects: 2022YFC3302500.

### Competing Interests
The authors declare that they have no competing interests.

### Author Contributions
- Feng Wen conceived and designed the experiments, performed the experiments, analyzed the data, performed the computation work, authored or reviewed drafts of the article, and approved the final draft.
- Zhuo Wang conceived and designed the experiments, performed the experiments, analyzed the data, performed the computation work, prepared figures and/or tables, authored or reviewed drafts of the article, and approved the final draft.
- Leda Qu performed the experiments, analyzed the data, performed the computation work, prepared figures and/or tables, authored or reviewed drafts of the article, and approved the final draft.

- Haixin Huang conceived and designed the experiments, performed the experiments, analyzed the data, performed the computation work, authored or reviewed drafts of the article, and approved the final draft.
- Xiaojie Hu conceived and designed the experiments, performed the experiments, analyzed the data, performed the computation work, prepared figures and/or tables, authored or reviewed drafts of the article, and approved the final draft.

## Data Availability

The raw data is available at figshare: 王，卓 (2023). row data.zip. figshare. Dataset. https://doi.org/10.6084/m9.figshare.23904213.v1.

## Supplemental Information

Supplemental information for this article can be found online at http://dx.doi.org/10.7717/peerj-cs.1962#supplemental-information.

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
