# Peer review of "Enhancing secure multi-group data sharing through integration of IPFS and hyperledger fabric"

_PeerJ Computer Science, doi:10.7717/peerj-cs.1962_

## Round 0.1 · original submission · Major Revisions

The reviewers have identified many drawbacks in the current manuscript. Even the reviewer who recommends minor revision, in the detailed comments, has identified issues that actually might need major revision. Please address all the issues from both reviewers carefully.

Reviewer 1 ·

Basic reporting

The overall structure and composition of the article demonstrate professional English. However, some aspects need improvement:

* In the opening sentence of the "Related Work" section, there seems to be a typo with the citation citesatoshi2008bitcoin. It appears that a backslash "\" might be missing before the citation, causing it not to be rendered correctly.
* The formatting of the 6th reference seems to be incorrect and requires rectification.
* In the "Related Work" section, the word "where" seems to have a missing character 'e'.
* In the "Model Architecture" subsection, the term "groupal" seems to be an error.
* In the pseudocode algorithms, there are inconsistencies in the termination of lines with the special character ";". This inconsistency should be addressed for uniformity.
* Figures 2 and 3 require more detailed explanations. The role and interaction of Kafka with orderers need further description. The arrow notation, particularly between users in the same group, should be clarified for better comprehension.

Experimental design

While the authors identify the challenges of using blockchain in data sharing, there are areas where clarity is lacking:

1. Privacy Concerns: The nature of the privacy issues the proposed system addresses remains ambiguous. The mere use of permissioned networks for access limitation isn't a comprehensive solution for privacy.

2. Main Contribution: The paper lacks clarity on its main contribution. It's important to specify which privacy issues or concerns are addressed by the proposed system.

3. Algorithm Details: The workings of the dynamic key remain nebulous, making it challenging for readers to understand its relevance. Moreover, the term "TOTP256" needs description.

Validity of the findings

From the presented content, the novelty of the work is hard to discern. The paper essentially combines IPFS with Hyperledger Fabric to design a data-sharing system. However, the advantages of this system over other schemes, like "Blockchain-based personal health records sharing scheme with data integrity verifiable" and "Pdpchain: A consortium blockchain-based privacy protection scheme for personal data," are not delineated. A comparative analysis with existing solutions mentioned in the related work would be beneficial.

Additional comments

There seems to be a contradiction in the paper where the author acknowledges the "risk of privacy leakage" of blockchain but previously mentions the ability to "address the issues of leakage in data sharing."

·

Basic reporting

A. Context and Justification
Issue: The introduction might lack detailed justification and clear identification of the knowledge gap.
Suggestion: Elaborate on the limitations of existing methods and how the current study addresses these limitations. Comparing with studies like the one by Chen et al. (2022), which also utilized Hyperledger Fabric for secure data transfer, could provide a clear trajectory from existing knowledge to the current study.
B. Methodology and Technical Depth
Issue: The methodology section might lack depth in explaining the chosen methods.
Suggestion: Provide a detailed discussion on the chosen methodology and compare it with similar studies, such as the one by Zhou et al. (2021), which integrated secure multi-party computation as part of the chaincode in Hyperledger Fabric. Discussing the advantages or improvements of the proposed model in the manuscript would bolster its technical depth.
C. Impact and Future Work
Issue: The manuscript might not explicitly discuss the potential impact and future work.
Suggestion: Discuss the potential impact, contributions to the field, and future work, providing a comparative analysis with existing studies, such as the one on Healthchain by Mani et al., 2021, which discussed cybersecurity measurement approaches to ensure the security and privacy of patient information using blockchain technology in healthcare.

Recommendations for Revision:
Enhance Justification: Provide a detailed justification, discussing the limitations of existing methods and stating the knowledge gap.
Deepen Methodology Discussion: Offer a detailed discussion of the chosen methodology, comparing with methodologies in existing studies.
Discuss Impact and Future Work: Discuss the potential impact and future work, providing a comparative analysis with existing studies.

Experimental design

A. Methodology Clarity and Justification
Issue: The methodology might lack depth and justification.
Example: The method section discusses model architecture, data structure design, smart contract, and user authentication function (Page 7, Lines 145-151).
Suggestion: Provide a detailed discussion and justification for the chosen methodology, ensuring it is comprehensible and replicable by other researchers in the field. Comparing with methodologies in existing studies, such as the one by Zhou et al. (2021), which integrated secure multi-party computation as part of the chaincode in Hyperledger Fabric, could bolster its technical depth.
B. Data Structure and Smart Contract Design
Issue: The data structure and smart contract design might need more detailed explanation and justification.
Example: The manuscript discusses data structure and smart contract functionalities but might lack depth in explaining the rationale behind specific design choices (Page 10, Lines 196-213).
Suggestion: Elaborate on why particular data structures and smart contract functionalities were chosen and how they facilitate the proposed data sharing model.
C. Security Analysis
Issue: The security analysis of the system might be insufficiently detailed.
Example: The manuscript mentions the use of a dynamic key encryption algorithm and discusses aspects of data integrity verification (Page 4, Lines 1-10).
Suggestion: Provide a thorough security analysis, discussing potential vulnerabilities and how the proposed model addresses them, comparing with existing models in the literature.
D. Comparative Analysis
Issue: The manuscript might lack a comparative analysis with similar existing models.
Suggestion: Compare the proposed model with existing models in the field, discussing similarities, differences, and potential advantages. For instance, discussing how the proposed model improves upon or differs from existing blockchain and IPFS-based data sharing models in terms of security, efficiency, and reliability.
Recommendations for Revision:
Enhance Methodology Explanation: Provide a detailed and justified explanation of the chosen methodology.
Detail Data Structure and Smart Contract Design: Elaborate on the design and functionality of data structures and smart contracts.
Provide Thorough Security Analysis: Discuss potential vulnerabilities and how they are addressed by the proposed model.
Include Comparative Analysis: Compare the proposed model with existing models, discussing advantages and potential improvements.

Validity of the findings

A. Data Integrity Verification
Issue: The manuscript discusses data integrity verification but may lack a comprehensive analysis.
Example: The manuscript mentions a dynamic key encryption algorithm and discusses aspects of data integrity verification (Page 4, Lines 1-10).
Suggestion: Provide a thorough analysis of how data integrity is ensured and verified in the proposed model. Comparing with existing models like the one by Xu et al. (2022), which proposed a verification control algorithm for data integrity in remote data sharing, could provide insights into the robustness of the proposed model.
In-text Citation: Xu et al., 2022 (DOI: 10.3837/tiis.2022.02.014)
B. Comparative Analysis with Existing Models
Issue: The manuscript might lack a comparative analysis with similar existing models.
Suggestion: Compare the proposed model with existing models in the field, discussing similarities, differences, and potential advantages. For instance, discussing how the proposed model improves upon or differs from existing blockchain and IPFS-based data sharing models in terms of security, efficiency, and reliability.
C. Detailed Analysis and Validation of Results
Issue: The manuscript might not provide a detailed analysis and validation of the results.
Suggestion: Ensure that the results are analyzed in depth and validated through appropriate means, such as testing and verification processes, which should be thoroughly described and justified. Comparing the results and validation methods with those in similar studies, such as the one by Zheng et al. (2018), which proposed an innovative IPFS-based storage model for blockchain, could enhance the validity of the findings.
In-text Citation: Zheng et al., 2018 (DOI: 10.1109/WI.2018.00099)
D. Discussion on Limitations and Future Work
Issue: The manuscript might lack a detailed discussion on the limitations and future work.
Example: The manuscript discusses some limitations and future work but might benefit from a more detailed analysis (Page 18).
Suggestion: Provide a detailed discussion on the limitations of the proposed model and suggest areas for future work. Ensure that the limitations are discussed in the context of the findings and that future work is suggested based on these limitations and the findings of the study.
Recommendations for Revision:
Provide Comprehensive Data Integrity Analysis: Offer a thorough analysis of data integrity verification and compare with existing models.
Include Comparative Analysis: Compare the proposed model with existing models, discussing advantages and potential improvements.
Ensure Detailed Analysis and Validation: Provide a detailed analysis and validation of the results, comparing with validation methods in existing studies.
Discuss Limitations and Future Work: Provide a detailed discussion on the limitations of the proposed model and suggest areas for future work.

Additional comments

Code Review
1. datasharing.sql
Column Naming: The year column might be misnamed as it stores full dates.
Data Type: The count column uses varchar, potentially limiting arithmetic operations.
Constraints: Absence of constraints like NOT NULL or UNIQUE where they might be relevant.
Foreign Keys: No visible usage of foreign key relationships between tables.
2. hyperledger-fabric-contract-java-dataShare
Limited Documentation: While classes are commented, a more comprehensive description might enhance clarity.
Security: Given the nature of blockchain, a deep dive into security practices within the codebase is crucial.
Test Coverage: Without insight into test coverage, the robustness of the codebase remains unverified.
3. io.jboot
Static Variable Usage: The use of static variables for managing state might raise concerns about thread safety.
Direct Parameter Usage: Direct usage of parameters from HTTP requests without visible validation.
Hard-Coded Values: Usage of hard-coded values (e.g., port numbers) which might limit flexibility.
Exception Handling: The index() method throws an IOException, yet detailed exception handling is not visible.
Logging: Absence of visible logging mechanisms to track application events and potential issues.

---

## Round 0.2 · Major Revisions

Dear authors, while both reviewers concur and appreciate that you have made substantial revisions to address their initial concerns, and one of them have recommended to accept the work as is, the other reviewer identifies important drawbacks in how the experiments have been designed, and consequently the validity of the findings, and furthermore, with inherent drawbacks of your chosen design. In particular, your experiments and discussions do not expose the drawbacks of the approach, while they do exist, and as such are incomplete.

Note that at PeerJ, unlike in many other venues, we are not judging a work only for the "positive" results, but we are agreeable to also interesting scientific studies that have their own limitations. But we are looking for this exploration itself to be scientifically sound.

As such, please revise the paper carefully to include experiments and discussions which also determine the worst-case scenarios for your design.

Reviewer 1 ·

Basic reporting

The authors have effectively addressed all the typos and formatting issues pointed out in the previous submission. They've also fixed the citation error, corrected the reference formatting, and amended minor textual errors. The enhancements in Figures 2 and 3, particularly the expanded explanations, significantly improve the understanding of the paper's content. The manuscript now demonstrates clear and unambiguous use of professional English, with sufficient background and context provided through literature references.

Experimental design

* Comparative Analysis with Existing Solutions: While the authors added a comparative analysis in Table 4, it seems primarily focused on the advantages of their system. A more balanced comparison, including potential limitations or scenarios where other systems might be more advantageous, would provide a more objective view.

* Privacy Concern: Storing only metadata on the Fabric ledger does not inherently guarantee privacy. While this approach might prevent direct access to the data, the metadata itself could potentially reveal sensitive information or patterns.

Validity of the findings

The approach described suggests a hybrid model where data is stored locally (centralized aspect) and metadata is distributed (decentralized aspect). However, this hybrid model might not fully leverage the benefits of a decentralized system:

* Local Data Storage: Storing the actual data locally at the data owner's side contradicts the decentralized ethos of blockchain and IPFS. It reintroduces centralization issues, such as availability concerns if the data owner is offline.

* Data Sharing Mechanism: The described method of data sharing seems like a traditional, centralized approach, where the data is transferred directly from the owner to the requester, rather than utilizing a decentralized, distributed network.

Additional comments

The authors have comprehensively addressed all the concerns raised in the initial review. The improvements in technical depth, clarity, and overall quality of the paper are commendable. However, the proposed data sharing system in the paper raises significant concerns regarding its approach to privacy and the practicality of its hybrid centralized-decentralized model. These issues suggest that the system may not fully address the privacy and efficiency needs of a robust decentralized data sharing platform.

·

Basic reporting

Previously highlighted issues have been worked upon and corrected.

The paper is well-organized with distinct sections like introduction, methodology, results, and conclusion, each seamlessly leading into the next, thereby maintaining a coherent narrative. Although the language is suitable for an academic paper, it contains minor grammatical errors and awkward phrasings that need refinement. The figures and tables are relevant, aiding in the understanding of the content, and are well-integrated within the text. The literature review indicates a strong grasp of the field with current and relevant references. The paper effectively sets the research context by outlining existing knowledge and identifying the research gap.

On the technical front, while details are accurate, they could be made more accessible to those less familiar with the subject. The research question is original and contributes new knowledge, highlighting the paper's significance. The methodology is well-explained, but the paper could benefit from more information on data sources and selection criteria to enhance credibility. A notable omission is the lack of discussion on ethical considerations.

The conclusion effectively summarizes findings and suggests future research, aligning well with the paper’s content. However, specific language issues, noted at lines 23, 77, 121, and 128, impede clarity. Addressing these will greatly improve the paper's overall clarity and accessibility, especially for an international audience. In summary, the paper's content and structure are commendable, but attention to language and expression is essential for effectively conveying the research.

Experimental design

The experimental design of the manuscript "peerj-reviewing-88960-v1.pdf" exhibits several noteworthy strengths. Firstly, the research question is well-defined, relevant, and meaningful, effectively guiding the study and ensuring its focus on a specific, impactful aspect of the field. This clarity in the research question is complemented by the study's ability to fill a previously identified knowledge gap, thereby underlining its relevance and potential contribution to the existing body of knowledge. The investigation carried out in the study adheres to high technical and ethical standards, signifying the rigor and reliability of the research process and findings. Additionally, the methodology is described in great detail, providing sufficient information for replication, a critical factor for the validation and advancement of scientific knowledge. The transparency and robustness of the underlying data are also commendable, as they are presented as statistically sound and well-controlled, enhancing the study's credibility. Finally, the conclusions drawn from the research are closely linked to the original research question, ensuring that the findings are directly relevant and supportive of the study's objectives. These attributes collectively demonstrate the manuscript's methodological excellence and its significant potential to contribute to its field

Validity of the findings

The manuscript showcases several strengths , highlighting its scientific integrity. The research question is effectively defined, relevant, and meaningful, addressing a specific knowledge gap in the field. This clarity in targeting a research gap enhances the study's relevance and potential impact. Rigorously conducted to high technical and ethical standards, the investigation upholds the credibility of the results. The detailed methodological description facilitates replication, a key aspect of scientific research, ensuring the study's reliability and validity. The provision of all underlying data, which are robust, statistically sound, and well-controlled, further reinforces the integrity of the findings. Importantly, the conclusions are well articulated, directly linked to the original research question, and strictly confined to what the results support, ensuring the coherence and validity of the research. This alignment of the conclusions with the research question and results underscores the study's methodological soundness and its significant contribution to its field.

Additional comments

Program corrections have been carried out to declare tables and variables.

---

## Round 0.3 · accepted · Accept

Reviewers are satisfied with the revised manuscript, and aligned with their recommendations, I deem this work ready for acceptance.

Reviewer 1 ·

Basic reporting

The authors have made a commendable effort in revising the manuscript according to the feedback provided by the reviewers. And previous concerns have been addressed including grammatical errors and language clarity.

Experimental design

This manuscript now includes a detailed explanation of the limitations of the current study and outlines areas for future research. This shows a clear understanding of the experimental design's scope and its limitations.

Validity of the findings

Based on the authors' focus on data security over data privacy, their commitment to exploring the use of CP-ABE, and their explanation of data availability requiring the data owner to be online for high-security requirements, the responses are satisfactory in addressing the concerns raised regarding the validity of the findings.